# Balance Rehabilitation through Robot-Assisted Gait Training in Post-Stroke Patients: A Systematic Review and Meta-Analysis

**DOI:** 10.3390/brainsci13010092

**Published:** 2023-01-03

**Authors:** Alberto Loro, Margherita Beatrice Borg, Marco Battaglia, Angelo Paolo Amico, Roberto Antenucci, Paolo Benanti, Michele Bertoni, Luciano Bissolotti, Paolo Boldrini, Donatella Bonaiuti, Thomas Bowman, Marianna Capecci, Enrico Castelli, Loredana Cavalli, Nicoletta Cinone, Lucia Cosenza, Rita Di Censo, Giuseppina Di Stefano, Francesco Draicchio, Vincenzo Falabella, Mirko Filippetti, Silvia Galeri, Francesca Gimigliano, Mauro Grigioni, Marco Invernizzi, Johanna Jonsdottir, Carmelo Lentino, Perla Massai, Stefano Mazzoleni, Stefano Mazzon, Franco Molteni, Sandra Morelli, Giovanni Morone, Antonio Nardone, Daniele Panzeri, Maurizio Petrarca, Federico Posteraro, Andrea Santamato, Lorenza Scotti, Michele Senatore, Stefania Spina, Elisa Taglione, Giuseppe Turchetti, Valentina Varalta, Alessandro Picelli, Alessio Baricich

**Affiliations:** 1Department of Health Sciences, Università del Piemonte Orientale “Amedeo Avogadro”, 28100 Novara, Italy; 2Physical Medicine and Rehabilitation Unit, AOU Maggiore della Carità University Hospital, 28100 Novara, Italy; 3Physical Medicine and Rehabilitation Unit, Polyclinic of Bari, 70124 Bari, Italy; 4Rehabilitation Unit, Castel San Giovanni Hospital, 29015 Piacenza, Italy; 5Theology Department, Pontifical Gregorian University, 00187 Rome, Italy; 6Physical Medicine and Rehabilitation, ASST Sette Laghi, 21100 Varese, Italy; 7Casa di Cura Domus Salutis, Fondazione Teresa Camplani, 25100 Brescia, Italy; 8Robotic Rehabilitation Section, Italian Society of Physical and Rehabilitative Medicine (SIMFER), 00187 Rome, Italy; 9Neurorehabilitation Department, IRCCS Fondazione Don Carlo Gnocchi, 20148 Milan, Italy; 10Experimental and Clinic Medicine Department, Università Politecnica delle Marche (UNIVPM), 60126 Ancona, Italy; 11Neurorehabilitation Unit, Bambino Gesù Children’s Hospital, 00165 Rome, Italy; 12Physical Medicine and Rehabilitation Unit, Centro Giusti, 50125 Florence, Italy; 13Unit of Spasticity and Movement Disorders, Division of Physical Medicine and Rehabilitation, University Hospital of Foggia, 71100 Foggia, Italy; 14Rehabilitation Unit, Department of Rehabilitation, “Santi Antonio e Biagio e Cesare Arrigo” National Hospital, 15122 Alessandria, Italy; 15Unit of Neurorehabilitation, Department of Neuroscience, Biomedicine, and Movement Sciences, University Hospital of Verona, University of Verona, 37126 Verona, Italy; 16Dipartimento Medicina, Epidemiologia, Igiene del Lavoro e Ambientale, Istituto Nazionale Assicurazione Infortuni sul Lavoro (INAIL), 00192 Rome, Italy; 17Italian Federation of Persons with Spinal Cord Injuries (FISH), 00197 Rome, Italy; 18Department of Physical and Mental Health and Prevention Medicine, Luigi Vanvitelli University of Campania, 81100 Naples, Italy; 19Department of New Technologies in Public Healthcare, Italian National Institute of Health (ISS), 00161 Rome, Italy; 20Translational Medicine, Dipartimento Attività Integrate Ricerca e Innovazione (DAIRI), Azienda Ospedaliera Santi Antonio e Biagio e Cesare Arrigo, 15122 Alessandria, Italy; 21Rehabilitation Unit, Santa Corona Hospital, 17027 Pietra Ligure, Italy; 22Tuscany Rehabilitation Clinic, 52025 Montevarchi, Italy; 23Department of Electrical Engineering and Information Technology, Polytechnic University of Bari, 70126 Bari, Italy; 24The BioRobotics Institute, Scuola Superiore Sant’Anna, 56025 Pontedera, Italy; 25Azienda Unità Locale Socio Sanitaria Euganea (AULSS 6), 35100 Padua, Italy; 26Rehabilitation Department, Valduce Villa Beretta Hospital, 23845 Costa Masnaga, Italy; 27Neurorehabilitation Unit, Santa Lucia Foundation IRCCS, 00179 Rome, Italy; 28Pediatric, Diagnostical and Clinical-Surgical Sciences Department, University of Pavia, 27100 Pavia, Italy; 29Neurorehabilitation Unit, Istituto Clinico-Scientifico Maugeri SPA IRCCS, 27100 Pavia, Italy; 30Pediatric Rehabilitation Unit, IRCCS Eugenio Medea, 23842 Bosisio Parini, Italy; 31Rehabilitation Unit, Versilia Hospital, 55041 Camaiore, Italy; 32Department of Translational Medicine, Università del Piemonte Orientale “Amedeo Avogadro”, 28100 Novara, Italy; 33Italian Association of Occupational Therapists (AITO), 00136 Rome, Italy; 34Rehabilitation Unit, Istituto Nazionale Assicurazione Infortuni sul Lavoro (INAIL), 56048 Volterra, Italy; 35Institute of Management, Scuola Superiore Sant’Anna, 56127 Pisa, Italy

**Keywords:** rehabilitation, robotics, balance, stroke, gait

## Abstract

Background: Balance impairment is a common disability in post-stroke survivors, leading to reduced mobility and increased fall risk. Robotic gait training (RAGT) is largely used, along with traditional training. There is, however, no strong evidence about RAGT superiority, especially on balance. This study aims to determine RAGT efficacy on balance of post-stroke survivors. Methods: PubMed, Cochrane Library, and PeDRO databases were investigated. Randomized clinical trials evaluating RAGT efficacy on post-stroke survivor balance with Berg Balance Scale (BBS) or Timed Up and Go test (TUG) were searched. Meta-regression analyses were performed, considering weekly sessions, single-session duration, and robotic device used. Results: A total of 18 trials have been included. BBS pre-post treatment mean difference is higher in RAGT-treated patients, with a pMD of 2.17 (95% CI 0.79; 3.55). TUG pre-post mean difference is in favor of RAGT, but not statistically, with a pMD of −0.62 (95%CI − 3.66; 2.43). Meta-regression analyses showed no relevant association, except for TUG and treatment duration (β = −1.019, 95% CI − 1.827; −0.210, *p*-value = 0.0135). Conclusions: RAGT efficacy is equal to traditional therapy, while the combination of the two seems to lead to better outcomes than each individually performed. Robot-assisted balance training should be the focus of experimentation in the following years, given the great results in the first available trials. Given the massive heterogeneity of included patients, trials with more strict inclusion criteria (especially time from stroke) must be performed to finally define if and when RAGT is superior to traditional therapy.

## 1. Introduction

Stroke is the second leading cause of death and the first cause of disability in the world. Given the rising mean age of the world’s population, the incidence of stroke is also steadily increasing, leading to higher economic burden through the years [1,2,3,4].

Aphasia and depression are often the causes of disability in post-stroke patients, but the most common disability tract is a reduced mobility due to hemiparesis [1,5]. This impairment is strictly connected to two main factors: a rapid change in muscle thickness and composition (visible within 1 month from the onset) and reduced central motor control [6,7]. All these impairments also lead to a reduced balance function: the patient is unable to maintain his center of gravity within his support base with or without the action of external forces [8]. After the clinical stabilization of the acute stroke, the rehabilitation program is usually focused on upper limb and gait rehabilitation, while balance is often a secondary or ignored outcome. This is a problematic point of rehabilitation programs because balance is a skill deeply involved in patients’ autonomy and independence. As a matter of fact, balance is not only fundamental in walking but also in many activities of daily life (ADL). It is also the main deficit involved in fall risk evaluation [9].

Nowadays, it is well known that an early and intensive mobilization protocol improves functional recovery after stroke, especially in young patients [10]. Along with the development of acute stroke management (early rehabilitation in stroke units, facilitation of brain repair mechanisms and experimental approaches, such as neuronal transplantation) [11], the management and rehabilitation of chronic stroke patients have recently seen great improvement. The greatest effort in stroke rehabilitation can be identified in four main outcome domains: physical fitness, ADL, arm-hand dexterity and function, and gait and mobility-related functions and activities [12]. Regardless of the main outcome pursued, many different neurorehabilitation techniques have been developed over the years for both sub-acute and chronic stroke survivors. Functional electrical stimulation (FES) has been largely used to contract muscles that are non-activable by the patient, improving complex motor function, such as gait. Along with peripheral stimulation, brain stimulation has been developed over the years. Transcranial direct current stimulation (tDCS) can be integrated during the rehabilitation program to improve limb movement and pain [13]. Manual therapy for strength and mobility recovery is the most used approach worldwide but is difficult to standardize, and it is heavily related to therapist dexterity and experience [14].

That is why, in the last 20 years, many technological devices have been developed, for both upper [15,16,17,18,19,20] and lower limb rehabilitation [21,22,23,24,25,26,27,28]. In this scenario, we can find many levels of assistance, with different ideas of rehabilitation beneath. First of all, there is the body weight-supported treadmill, in which the patients can practice a supervised and repetitive walk. Then, we can find overground exoskeleton, which provide patients with either full or partial guidance of the lower limbs during the whole gait cycle [13]. Lastly, there are also end-effectors, which are smaller devices that permit the patient to perform a specific joint movement during specific gait phases [29]. The development of these devices has improved the quality of post-stroke rehabilitation, guaranteeing patients early verticalization, gait training, and dismission [30,31,32].

The use of robotics for lower limb rehabilitation is currently starting to spread widely, but given the various protocols available worldwide, there is no consensus on which approach is the most effective [10]. Moreover, there is a lack of knowledge about the superiority of robotic treatment over standard treatment and which of the patients’ characteristics are to be considered when deciding whether or not to treat them with robotic devices. Furthermore, while there is some evidence on the efficacy of these devices for gait rehabilitation [33,34,35], more evidence on the balance outcomes is needed. More precisely, very few trials on robotic rehabilitation consider balance as an outcome and all the previous meta-analyses performed were not able to gather enough data to provide a sufficient statistical relevance. In addition, considering the high cost of these robotic devices, there is no certain evidence on their efficacy that completely justifies this kind of expenditure. The aim of this study is to sum up all the evidence about robotic-assisted gait training (RAGT) on balance rehabilitation. In particular, the study focuses on covering the lack of statistical relevance present in the actual literature, due to a small number of trials included, and tries to assess as many sub-group evaluations as possible.

## 2. Materials and Methods

### 2.1. Selection Criteria

PubMed, Cochrane Library, and PeDRO electronic databases were searched; the last online search was performed on 3 August 2022. Only randomized clinical trials (RCT) with full text available in English or Italian have been considered eligible. All articles published up to June 2022 were considered. The systematic online research was performed using the following MeSH terms: “stroke”, “postural balance,” “robotics,” “rehabilitation” AND string “robotics [mh]” OR “robot-assisted” OR “electromechanical” AND “rehabilitation [mh]” OR “training” AND “postural balance [mh]”.

This research has been conducted following the PRISMA 2020 guidelines. The meta-analysis has been registered on PROSPERO (ID CRD42022348043).

### 2.2. Inclusion and Exclusion Criteria

Studies were included in the meta-analysis if: (1) they were randomized controlled trials; (2) the participants’ age was ≥18 years; (3) they included post-stroke survivors only; (4) the intervention was any kind of RAGT (robotic gait training on treadmill, overground exoskeleton, or end-effector); (5) the control group comprised of conventional treatment (physical exercises, non-robotic overground, or treadmill walking training); (6) balance evaluation was a primary or secondary outcome using Berg Balance Scale (BBS), Timed Up and Go test (TUG), or both; and (7) they reported the average of the outcomes measured at the baseline and at the end of follow-up or the difference between the two measures, the corresponding standard deviation (SD), and the number of subjects randomized to each treatment arm. Studies considering participants with a history of multiple strokes or using RAGT in both experimental and control groups were excluded from the study.

Studies’ eligibilities were independently assessed by six authors; disagreements between operators were discussed with the study coordinators who decided whether or not to include the study. The reference lists of the included studies were also screened to identify papers potentially missed by the literature search.

### 2.3. Quality Assessment

The revised Cochrane risk-of-bias tool for randomized trials (RoB 2) was used to assess the risk of bias associated to each experimental study included in the review by four authors independently. If there was no consensus between the two operators, the supervisor was consulted to decide about the assessment.

### 2.4. Data Extraction

The following information was extracted from the included articles: first author’s last name, year of publication, number of participants treated as the experimental and control group, robotic device used in the experimental group, time from stroke, protocol applied in the experimental and control group, single-session duration, number of sessions per week, weeks of treatment, BBS and TUG values pre- and post-treatment for both experimental and control group, and p-values referring to the comparison between the two groups.

### 2.5. Statistical Analysis

DerSimonian and Laird method was used to calculate the random effect pooled mean differences (pMD) and the corresponding 95% confidence interval (95% CI) of the changes in the outcome measures between the end of follow-up and the baseline in the treatment (RAGT) and control groups (conventional treatment). Random effect model was chosen since between-studies heterogeneity was expected due to systematic differences in the devices considered and in the way the studies were conducted (e.g., follow-up duration, number of sessions among others). To perform the analyses, firstly, the change in outcome measures between the end of follow-up and the baseline was calculated as the difference in the means measured at the two time points separately for the intervention and control groups. Thereafter, for each pre-post change, its standard deviation (*SD*) was calculated as follows:(1)$$\sqrt{var_{pre}+var_{post}-2\times \rho \times SD_{pre}+SD_{post}}$$
where *var_pre_* is the variance of the mean of outcome measured at the baseline, *var_post_* is the variance of the mean of outcome measured at the end of follow-up, and *SD_pre_* and *SD_post_* are the corresponding standard deviations. Moreover, *ρ* is the correlation coefficient between pre- and post-measurements, and it was set to 0.6 (based on gathered data). Then, the differences in the changes between intervention and control groups (MD) were calculated as well as the corresponding *SD* (square root of the sum of the variances of the two changes). The statistical analysis has been performed separately for BBS and TUG values. Between-studies heterogeneity was assessed both with the Cochran’s Q homogeneity test and the I2 index. The values of the index greater than 50% are suggestive of between-studies heterogeneity. For multi-arm trials, the intervention associated with the lowest treatment effect was selected to obtain more conservative pMD and to avoid within-study correlation of treatment effects. 

Moreover, random effect meta-regression models were applied to assess the impact of selected treatment’s characteristics on pMD, namely treatment duration, weekly sessions, duration of each session, and robotic device used (given gathered data and the machine utilization in the studies, information summarized as Lokomat^®^ (Hocoma, Volketswil, Switzerland) vs. others robotic devices).

Supplemental analyses were also performed. Firstly, a sensitivity analysis was performed calculating the pMD including the estimates of the arm with highest treatment effect for multi-arm trials to assess the potential change in the treatment effect due to the chosen arm. Secondly, an influence analysis was conducted in order to evaluate if the pMD estimate was strongly influenced by single studies. The procedure consists of calculating the pMD, excluding one study at a time, and evaluating how much pMD varies from its original value. Publication bias was visually assessed using funnel plot for each intervention–outcome combination studied and tested using Egger’s test. All analyses were performed with R version 3.4.4 (package “meta”, R Foundation for Statistical Computing, Vienna, Austria). 

## 3. Results

### 3.1. Literature Selection Process

Figure 1 shows the flow diagram of the study selection strategy. In the first search draft, 1522 articles were found. After excluding all recurring search results, 719 articles remained. Of these, 422 studies were excluded because they were not pertinent with the present research, not RCT-studies, or written in other languages than English or Italian. Of the 297 remaining articles, based on their abstract, 207 were excluded because they were not RCT studies or did not evaluate the outcomes considered in this meta-analysis. The remaining 90 were evaluated on their full text. Of these 90 studies, 71 were excluded because the full text was unavailable or the protocol did not fully respect the inclusion and exclusion criteria. At the end of the decision process, 19 studies were included in the meta-analysis. One of these studies, even meeting all the criteria, did not present all the values necessary to assess pMD. After writing to the authors but receiving no response, the study was excluded from the analysis [36].

Appendix A shows all the data extracted from the selected studies. A total of nine studies selected only BBS as the outcome [37,38,39,40,41,42,43,44,45], three only used TUG [46,47,48], while six have considered both [49,50,51,52,53,54]. Regarding the robotic devices used in the studies, seven performed the trial with Lokomat^®^ [40,43,44,45,51,52,53], while the others decided to use other devices, both commercialized and prototype.

Focusing on the studies by Mustafaoğlu [53], Park [51], and Yeung [41], which present two experimental groups and one control group, in the first trial, we have chosen the experimental group treated with the combination of RAGT and conventional therapy, while for the second and third study, RAGT + auditory stimulation and power-assisted ankle robot, respectively (experimental groups with lowest treatment effect).

Regarding quality assessment, all included trials showed a sufficient quality level in the initial protocol and during experimentations development, with a moderate risk of bias. The absence of double and triple blinding protocols is intrinsic to this topic. In fact, using one tool or machine over conventional manual treatment cannot be blinded. However, this does not represent an important bias, because all considered outcomes are machine-registered; knowing the kind of rehabilitation performed does not modify the evaluation.

### 3.2. Primary Analyses

Figure 2 shows the forest plot regarding BBS reporting the study’s specific mean differences and the corresponding 95% CI, pMD, and heterogeneity evaluation. The results show that pre-post mean difference in BBS score is higher in patients treated with RAGT than conventional therapy with a pMD of 2.17 (95% CI 0.79; 3.55), suggesting a higher improvement in BBS in the RAGT group compared to conventional therapy. Heterogeneity is not particularly marked, with an I^2^ index of 46%, but the homogeneity hypothesis is still rejected (*p* = 0.03). Changing the considered experimental group for the Mustafaoğlu [53], Park [51], and Yeung [42] studies, the pMD result is 2.31 (95% CI 0.75; 3.87). The result shows a better outcome for RAGT treatment, but substantially equal to the previous pMD. On the other hand, heterogeneity increases considerably (I^2^ index equal to 62.2%, *p* = 0.0007). Changing the considered experimental group for the Park [51] and Yeung [41] studies, the pMD result is 2.31 (95% CI 0.75; 3.87). The result shows a better outcome for RAGT treatment, but substantially equal to the previous pMD. On the other hand, heterogeneity increases considerably (I^2^ index equal to 62.2%, *p* = 0.0007). 

Figure 3 shows the forest plot regarding TUG values. Similar to the previous analysis, pre-post mean differences of TUG are in favor of RAGT treatment rather than conventional treatment. In this case, however, the result is not statistically relevant, with a pMD of −0.62 (95% CI − 3.66; 2.43). A strong heterogeneity between studies has been detected (I^2^ index = 74%, *p* < 0.01). When the experimental group of the Mustafaoğlu [53] and Park [51] studies were substituted, the efficacy of RAGT slightly increases, with a pMD of −1.09 (95% CI − 4.53; 2.35), but still remains non-statistically relevant. As shown above, heterogeneity also increases (I^2^ index = 81.9%, *p* < 0.0001).

### 3.3. Supplemental Analyses

Table 1 shows the results of the meta-regression performed, showing beta values along with its 95% CI and *p*-value for each analysis for both BBS and TUG. None of the considered variables seem to correlate with BBS improvement. Considering TUG, there is a direct proportion between TUG improvement and total rehabilitation duration: the longer the treatment lasts, the more TUG pMD values for TUG decreases, showing an improvement among patients (β = −1.019, 95% CI − 1.827; −0.210, *p*-value = 0.0135).

Figure 4 reports the results obtained by the influence analyses. The graph shows the pMD obtained omitting the study reported on the left part of the Figure. Influence analysis has been conducted for both BBS and TUG evaluation. In both cases, all the partial pMD calculated are included within the total pMD 95% CI. Hence, there is no study that influences relevantly the meta-analysis performed.

Lastly, Figure 5 reports the results of publication bias analyses for BBS and TUG evaluation. The funnel plot visually confirms the symmetry of studies outcomes when compared to pMD in both evaluations, excluding the presence of publication bias. Egger’s test also confirms this affirmation, with a *p*-value = 0.8519 for BBS and *p*-value = 0.9323 for TUG.

## 4. Discussion

The present meta-analysis demonstrates how in the last 20 years robotic rehabilitation has spread widely. In most cases, however, RAGT is only focused on gait rehabilitation, often neglecting balance deficits and their treatment. The few trials that directly evaluate balance mostly consider BBS, but this score mostly considers balance in a static environment or, in the few dynamic evaluations, it is based on a single task at time. TUG test, based on how it is performed, can show how balance is maintained in a more complex environment, with different predicted disturbances to contrast [55]. Moreover, even though it has a great reliability, BBS might not detect modest but clinically important changes in balance [56]. Considering both BBS and TUG, therefore, is a good and reliable combination of static and dynamic balance evaluation.

Almost all the included studies agreed on RAGT efficacy on balance when compared to baseline data. This is not only statistically relevant, but also clinically relevant, given that nearly all the studies present BBS pre-post change greater than the minimal change for clinical improvement [57]. On the other hand, however, about half of the studies included demonstrated the superiority of robotic treatment over conventional rehabilitation in BBS or TUG. In many cases, the trials showed that RAGT and conventional rehabilitation improved balance in stroke patients, with no relevant differences between groups [39,41,42,43,46,50,53,54].

The limited efficacy of RAGT can be related to many aspects. First of all, as highlighted by Zhang et al. [58] in their meta-analysis, there are many different robotic devices available in commerce or under investigation. This profound heterogeneity in robotic devices, even if not drastically, surely is a confounder when comparing the outcomes between studies. Another similar concern is the great differences in treatment protocols between studies. In fact, RAGT was very variable, with a minimum of 2 [41,42,54] and a maximum of 20 weeks of treatment [52]. Session frequency was also very variable, with a weekly number of sessions between 2 and 7, and each session lasting from 30 to 120 min. Last but not least for importance is the profound heterogeneity among included stroke survivors between studies. Most studies reported the time since stroke as open classes (e.g., ≥3 months, ≥6 months, ≥12 months, <12 months), preventing any possible meta-regression based on this characteristic. Even if all protocols agreed on including only first-stroke patients, with a sufficient cognitive capacity to execute operator commands and perform the therapy, other characteristics were very variable. For example, only a few studies have considered the Functional Ambulation Classification (FAC) as a criterion, and, when considered, the range included in the study was extremely wide, usually FAC 3 or higher. Time from stroke was also an extremely variable criterion. Most studies selected both sub-acute and chronic patients, usually with a lower limit of exclusion but not an upper limit. Age was also extremely variable: the only limit usually set was an age ≥ 18 years, without any upper restriction. This is a great confounder, given that only 10% of acute stroke occurs in people 45 years old or younger [59], but they are also the patients with the best probable outcome [60,61].

All these variables lead to very different pathological cases, and a substantial non-superiority of RAGT over traditional treatment in trials. This is an extremely relevant issue, which is also highlighted by Lorusso et al. [62] in their recent systematic review. Creating trials more stratified by remaining autonomy, age, and time from acute stroke is mandatory in order to fully understand the potential of RAGT. We can affirm, however, that the combination of RAGT and conventional treatment seems the most effective protocol in post-stroke patients. In the majority of the selected studies, greater statistically relevant changes are detected after training with both techniques, while using only RAGT is basically equal to conventional treatment alone. This affirmation is particularly highlighted in the trial by Mustafaoğlu [53], in which the combination of RAGT and conventional treatment has been described as far superior to the two techniques applied singularly. This concept is added to, as confirmed by this meta-analysis, the model that a longer protocol is fundamental for regaining a complex task such as balance. Stability rehabilitation, in fact, is a mix of sensibility, muscle strength, and coordination [13,63]: in order to recover all these abilities, a large rehabilitation window is needed.

Another important aspect in favor of RAGT is a major impact on patient neuroplasticity. Many authors have demonstrated the large implication of neuroplasticity in robotic rehabilitation of the upper limb [64,65]. This superior effect could theoretically translate to lower limb robotic rehabilitation: a first example is shown in the trial by Kim et al. [66], which showed an improved neuroplastic modulation during RAGT using end-effector robotics. The mechanisms involved in neuroplasticity, however, are still unclear, and a correlation between robotic rehabilitation and improved neuroplasticity is yet to be found. This is highlighted by Kim et al., who stated that no clear superiority over traditional treatment has been found [66]. Moreover, there are some studies that emphasize the improved neuroplasticity using non-robotic approaches, such as high intensity interval training and neurorestoration techniques [67,68]. Given all these findings, it is possible that the combined approach of RAGT and traditional treatment can lead to a better outcome adding the neuroplastic drive of both techniques. Once again, further studies need to be developed in order to better understand the mechanisms and the correlation between neuroplasticity and rehabilitation outcomes.

As technology develops, robotic rehabilitation also grows, creating new treatment possibilities, but with, at least for now, uncertain results. For example, different combinations of RAGT and other devices are starting to spread, such as with electrostimulation or biofeedback technologies, but an effective additional benefit is still controversial and more studies need to be conducted [51,69]. On the other hand, however, robotic devices (in particular exoskeletons) remain highly expensive. Even though some authors affirm that performing robotic rehabilitation has a minor economic impact than traditional rehabilitation [70], the required investment is, at the present, really high, impeding this technology to be available in every rehabilitation center, especially public clinics. Even though the association of the two techniques seems to be the better neurorehabilitation approach, it cannot fully justify this kind of cost. Therefore, along with the development of new technologies, an effort to reduce robotic costs must be made.

The most interesting results, however, are shown by trials that used balance-specific robotic devices. In these cases, stability recovery appears greater than the results obtained with RAGT [71,72,73,74]. In contrast with RAGT, which is an essential but very task-specific training, balance-based training represents a more wholesome approach that not only is fundamental in gait but also in almost all ADL, as described by Lorusso et al. [62]. The first trial’s results in this field are very noticeable and further studies could definitely validate this new uprising approach, also considering that robotic devices for balance are usually less expensive than the devices used in RAGT.

Even if this meta-analysis presents the largest sample of patients treated with RAGT and evaluating balance available in the current literature, it also presents several limitations. First of all, the high heterogeneity between studies is a great confounder of our results: very different protocols have been performed and defining with statistical certainty, which is the best, was not possible. In addition, there is no clear evidence on which RAGT protocol leads to a better balance improvement. Even if there is proof that the rehabilitation total duration leads to a better outcome for dynamic balance, there is still no correlation between outcome rehabilitation intensity and single-session duration. Moreover, the very different inclusion criteria of each study, especially considering the time from stroke, were a strong limitation for more specific meta-regression. Not being able to perform a sub-group analysis considering acute, sub-acute, and chronic patients leaves a lack of knowledge in this field, and more trials considering precisely one of these categories must be conducted in the future. Finally, with the massive heterogeneity of robotic devices used in the considered trials, a sub-group analysis considering the type of robotic device used was not possible. Determining which device between robotic treadmill, overground exoskeleton, and robotic end-effector has the best outcome on balance of stroke survivors is still to be determined in the future. 

## 5. Conclusions

The present meta-analysis provides information on the efficacy of RAGT on balance of stroke survivors. Although some trials have reported a major efficacy, the statistical evaluation performed shows that RAGT does not reach better outcomes than traditional therapy. Even though there is no statistically clear evidence (due to the great heterogeneity between studies), it seems that the combination of RAGT and conventional treatment could lead to better outcomes than the two techniques performed singularly. Therefore, on a clinical level, RAGT and conventional treatment should be performed together when both available in the case of post-stroke patients with balance impairment.

RAGT is, per definition, gait-centered, and balance is almost always a secondary outcome. Given the results of recent trials on the topic, robot-assisted balance training should be more scientifically investigated in the following years, in order to rapidly insert it into the common clinical activities. For both RAGT and stability-specific robotic training, trials with more structured inclusion criteria must be performed in the future. Performing a sub-group analysis considering age and time from stroke could lead to a more patient-specific rehabilitation protocol and finally define whether robotic rehabilitation has a better outcome than conventional treatments and in which patients.

## Figures and Tables

**Figure 1 brainsci-13-00092-f001:**
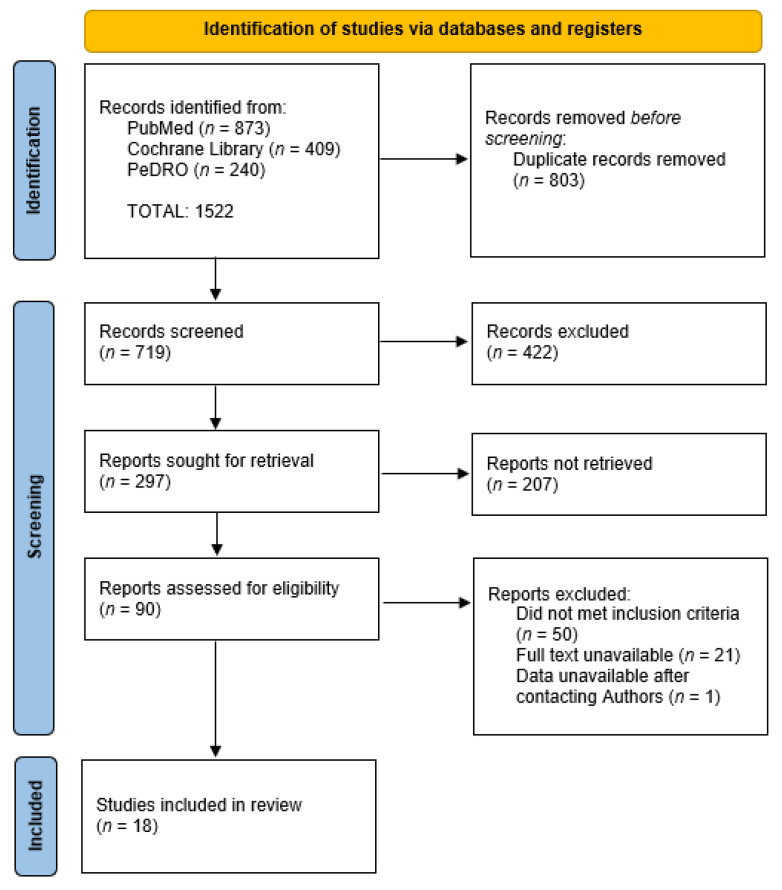
Flow diagram representing the study selection process for the meta-analysis.

**Figure 2 brainsci-13-00092-f002:**
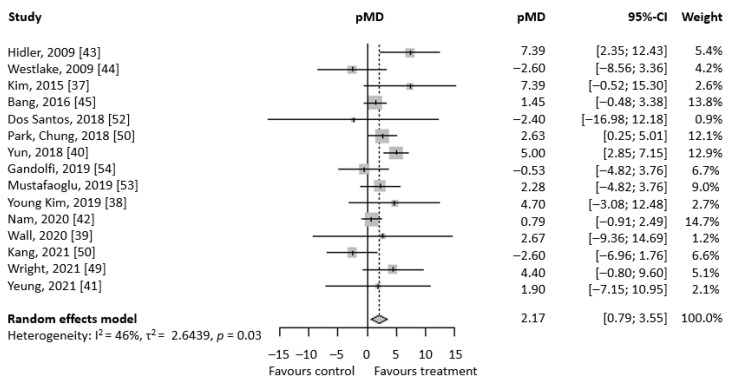
Forest plot regarding BBS meta-analysis. On the right, there are the mean pre-post difference of each selected study and their respective 95% CI. At the bottom, there are total pMD, its 95% CI, and heterogeneity evaluation.

**Figure 3 brainsci-13-00092-f003:**
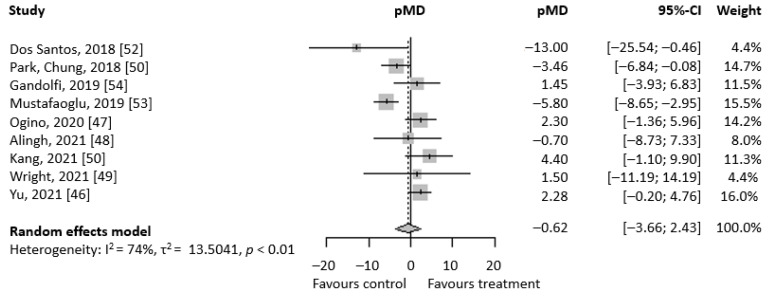
Forest plot regarding TUG meta-analysis. On the right, there are the mean pre-post difference of each selected study and their respective 95% CI. At the bottom, written in bold, there are total pMD, its 95% CI, and heterogeneity evaluation.

**Figure 4 brainsci-13-00092-f004:**
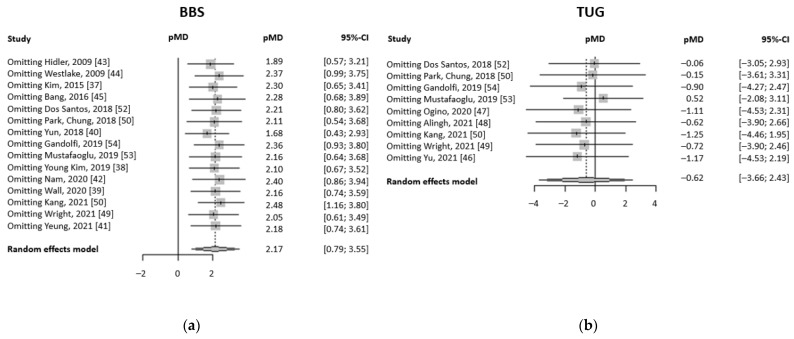
Forest plots summarizing the influence analyses performed: (**a**) pMD obtained omitting one study at a time (on the left) in BBS meta-analyses; (**b**) pMD obtained omitting one study at a time (on the left) in TUG meta-analyses.

**Figure 5 brainsci-13-00092-f005:**
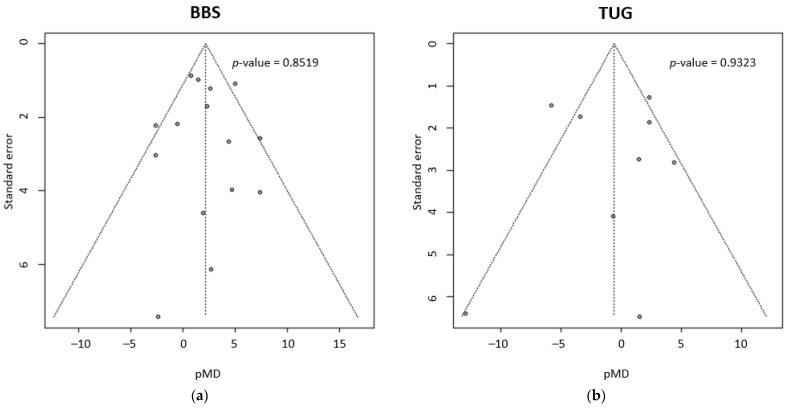
(**a**) Funnel plot that analyses potential publication bias in BBS meta-analysis. On the top right corner, *p*-value calculated with Egger’s test for publication bias. (**b**) Funnel plot that analyses potential publication bias in BBS meta-analysis. On the top right corner, *p*-value calculated with Egger’s test for publication bias.

**Table 1 brainsci-13-00092-t001:** Meta-regression performed on BBS and TUG. Each analysis is described with its β values and its respective 95% CI. * *p*-value < 0.05.

	BBS			TUG		
Variable Studied	β	95% CI	*p*-Value	β	95% CI	*p*-Value
Treatment duration (in weeks)	0.222	(−0.512; 0.957)	0.5529	−1.019	(−1.827; −0.210)	0.0135 *
Weekly sessions	1.074	(−0.626; 2.775)	0.2157	−1.333	(−3.197; 0.530)	0.1608
Single-session duration	0.051	(−0.043; 0.145)	0.2854	0.040	(−0.041; 0.121)	0.9726
Devices: Lokomat^®^ vs. others	0.873	(−2.572; 4.317)	0.6196	−3.566	(−7.629; 0.498)	0.0854

## Data Availability

The data presented in this study are available within the text and in the Appendix A.

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
