# Peer review of "Balance Rehabilitation through Robot-Assisted Gait Training in Post-Stroke Patients: A Systematic Review and Meta-Analysis"

_brainsci, 2023, doi:10.3390/brainsci13010092_

Round 1
Reviewer 1 Report
Comments and Suggestions for Authors
This systematic review and meta-analysis aimed to are to summarize systematically the available evidence about robotic gait training (RAGT) on balance rehabilitation post-stroke. After a comprehensive computerized search, the authors included 18 RCTs according to their inclusion criteria. They reported no superiority of RAGT over conventional intervention. The topic is interesting and summarizes important evidence. There is some minor comments listed below:
The abstract needs some improvement, and the result and conclusion are not clear. For example, the authors concluded that “RAGT is superimposable to traditional therapy. RAGT and conventional treatment combination seem to provide better outcomes than the two techniques singularly. Robot-assisted balance training should be defined in the following years. Trials with more structured inclusion criteria must be performed.” This conclusion is not clear, What the authors mean by robot-assisted balance training should be defined in the following year.
Introduction,
Lines 96-98, I agree with the authors that the functional electrical stimulation FES was widely used, however, tDCS is not electrical stimulation and could not compare with FES. It’s not true that FES and tDCS are used to facilitate the affected limb and its movement. This sentence should be modified to be clearer.
In the last paragraph of the discussion, the authors presented the rationale and purpose of this study. Indeed, the authors ignored presenting the cost of these devices compared to their real effectiveness. Please mention this point clearly.
Materials and Methods
Line 122, the databases were searched not investigated.
There is a typo in line 125 “Onile”.
Do the authors follow PRISMA guidelines, if yes please mention this appropriately.
The results were well-written and reported.
The discussion
The authors should discuss also the possible role of RAGT on neuroplasticity post-stroke.
The authors ignore the limitation of this study. I think there are a lot of limitations related to this review. For example, sub-group analysis was not performed, this is one of the main limitations. In addition, the effect of RAGT on acute, sub-acute, and chronic stroke are not clear, due to the high heterogeneity between the included studies. The optimal RAGT protocol to improve balance post-stroke is not clear as well. Please report any other limitation related to this study appropriately.
Another important aspect, the clinical recommendation for clinical practice and future research are missing. The authors should state a clear recommendation depending on the current results for both clinical practice and future research.
Though the text there are some grammatical errors, the English language should be improved.
Author Response
Dear Reviewer,
First of all, I would like to thank you for the time you have spent reading and evaluating our manuscript. Your comments are a great chance to improve our present and future work.
As you highlighted, we performed a critical proofread with an English native speaker, improving manuscript language and style. We also clarified all the issues you have found in the Abstract, Introduction, Material and Methods and Discussion section.
You may find a precise response to every issue in the file attached.
Best regards.

Reviewer 2 Report
Comments and Suggestions for Authors
The authors conduct a systematic review and meta-analysis to show the effect of RAGT on balance in stroke participants. The work in present form is not suitable for publication. My major concern is the inclusion criteria; it considers only two outcomes. Furthermore, there is no well justification for the study. Please find my following comments:
Revise Line 56
As the authors mentioned that ‘’ no strong evidence about RAGT superiority, especially on balance’’, why did you conduct a systematic review???
Why the included outcomes were only berg balance scale, and TuG? What is about posturgraphy and other balance outcomes measures??
Line 68 ‘’ superimposable’’?, The RAGT can be used as adjunctive intervention
Line 70 ‘’ Trials with more structured inclusion criteria must be performed ‘’, This statement is not correct
Line 78-81 is out of context
The introduction is not well written. There is no justification of the study aim. I think that there is at least 10 published meta-analyses about RAGT for balance in stroke.
Is the fixed-or random effect model was used in the pooled effect size analysis? And why?
Why the meta-regression did not consider time since stroke factor?
There is no clear conclusion
Author Response
Dear Reviewer,
First of all, I would like to thank you for the time you have spent reading and evaluating our manuscript. Your comments are a great chance to improve our present and future work.
As you highlighted, we performed a critical proofread with an English native speaker, improving manuscript language and style. We also clarified the issues you have found along the manuscript.
You may find a precise response to every issue in the file attached.
Best regards.

Reviewer 3 Report
Comments and Suggestions for Authors
The authors present a good work; however, novelty is the main concern that need to be clearly highlighted.
Kindly proofread whole content of this manuscript.
Author Response
Dear Reviewer,
First of all, I would like to thank you for the time you have spent reading and evaluating our manuscript.
As you have highighted in your comments, we performed an accurate manuscript proofread with an English native speaker, improving the language and style used in the manuscript.
We also have stated more clearly in the Introduction and Discussion sections the importance and the novelty of this meta-analysis.
You may find a more accurate description about the suggested revision in the attached file.
Best regards.

Round 2
Reviewer 2 Report
Comments and Suggestions for Authors
The concern is still on the novelty of the study. I recommend rejection for this manuscript